# Immunodominant B-Cell Linear Epitope on the VP1 P Domain of a Feline Norovirus Cat Model

**DOI:** 10.3390/pathogens11070731

**Published:** 2022-06-27

**Authors:** Tomomi Takano, Mizuki Ryu, Tomoyoshi Doki, Hajime Kusuhara

**Affiliations:** 1Laboratory of Veterinary Infectious Disease, School of Veterinary Medicine, Kitasato University, Towada 34-8628, Japan; vm16131@st.kitasato-u.ac.jp (M.R.); doki@vmas.kitasato-u.ac.jp (T.D.); 2Health and Environment Research Institute, Yokkaichi 512-1211, Japan; kusuhh00@pref.mie.lg.jp

**Keywords:** norovirus, B-cell epitope, surrogate model, cat

## Abstract

Norovirus (NoV) infection remains a major public health concern worldwide. Appropriate animal models are essential for the development of effective NoV vaccines. We previously established the feline NoV (FNoV)-cat model as a surrogate animal model for human NoV infection. In the present study, we analyzed the B-cell linear epitope in the P domain of FNoV to confirm the basic immunological features of the FNoV-cat model. B-cell linear epitopes were present in the P2 subdomain. We compared antibody levels to peptides containing the B-cell linear epitope (P-10) in three FNoV-infected cats with time-course changes in viral load and symptom scoring. After FNoV infection, viral shedding and clinical symptoms were shown to improve by elevated levels of antibodies against P-10 in the plasma. This report provides important information for understanding NoV infections in humans and cats.

## 1. Introduction

Human norovirus (human NoV; HNoV) is a major cause of acute gastroenteritis. HNoV infection occurs more frequently in younger generations, especially in children under five years of age [1]. Typical symptoms of HNoV infection are vomiting and watery diarrhea. Most patients recover within a few days, but those with underlying medical conditions, immunocompromised individuals, and elderly persons may develop severe symptoms. There are an estimated 684 million cases of HNoV infection and 0.2 million deaths every year [2]. HNoV infections are estimated to cost $60 billion due to social costs worldwide [3]. In veterinary medicine, animal noroviruses are recognized as potential pathogens of gastroenteritis in cats, dogs, and cows [4,5,6,7,8].

Noroviruses are members of the family *Caliciviridae*, order *Picornavirales*. The viral genome is a positive-sense, single-stranded RNA of approximately 7.5 kbp and is comprised of three open reading frames (ORFs) [9]. ORF1 encodes six nonstructural proteins including RNA-dependent RNA polymerase. ORF2 encodes the major capsid protein (VP1), and ORF3 encodes the minor capsid protein (VP2). VP1 can be divided into two major regions: the shell (S) domain and the protruding (P) domain [10]. The P region consists of the P1 subdomain and the P2 subdomain. The P2 subdomain binds to receptors for histo-blood group antigen (HBGA) entry into intestinal epithelial cells [11]. Noroviruses are classified into genogroup GI to GX based on differences in the amino acid sequence of VP1 [9]. For example, HNoVs are classified into six genogroups, GI, GII, GIV, GVII, GVIII, and GIX, whereas feline noroviruses (FNoVs) are classified into two genogroups, GIV and GVI [12]. Genetically, GIV and GVI FNoVs are located more closely to human noroviruses than mouse noroviruses (MNoV: GV) and bovine noroviruses (GIII). GIV FNoV has been identified in cats in the U.S. and Japan, and GVI FNoV in cats in Japan and Italy.

We previously showed gastrointestinal symptoms in SPF cats orally challenged with FNoV gene-positive feces. This fact strongly suggests that FNoV is a pathogen that causes gastroenteritis in cats. At present, FNoV infection has not attracted much clinical attention as a result of its low symptoms. However, NoV has a higher rate of genetic mutation than other viruses. Therefore, it is possible that FNoV may mutate in the future to become a highly virulent strain.

Animal models of infectious gastroenteritis caused by viral infection are necessary for elucidating the pathogenesis of NoV infection. There are no reports of non-human animals presenting gastroenteritis due to HNoV infection, except for wild-type neonatal mouse and gnotobiotic animals that have poorly developed immunity [13,14]. Adult mice and pigs do not develop gastroenteritis after inoculation with host-specific NoVs. On the other hand, cats inoculated with FNoV show vomiting and diarrhea [15]. Therefore, FNoV infection in cats may be a useful surrogate animal model for elucidating the pathogenesis of infectious gastroenteritis caused by HNoV in humans.

A better understanding of antibody-antigen interactions is important in vaccine development against NoVs. B-cell epitope mapping is an approach for identifying immunodominant epitopes of viruses [16]. B-cell epitopes are classified into linear and discontinuous epitopes [17]. Linear epitopes are stretches of continuous amino acid sequences. The placement of linear epitopes is identified using immunological analysis with short peptides (10–20 mer) [18]. The results obtained from such analysis provide a valuable profile for the development of vaccines. At present, the search for the B-cell linear epitope of NoV is mainly conducted using recombinant virus-like particles (VLPs) and VLP-derived monoclonal antibodies [19,20,21,22]. Anti-HNoV monoclonal antibodies purified from human blood are used in the epitope analysis of HNoV [23].

We reported that symptom scores and virus shedding in feces were inversely correlated with anti-recombinant partial VP1 (aa228-aa419) IgG levels in FNoV-infected cats [24]. Based on this fact, it is speculated that the immune response induced by the B-cell epitope in VP1 inhibits viral shedding and reduces symptoms of FNoV infection. However, B-cell linear epitope mapping of FNoV has not been performed to date. We postulated the presence of B-cell linear epitopes in the P domain of FNoV VP1 and prepared 20-mer overlapping linear peptides derived from the P domain. The reactivity of these peptides and GVI FNoV-infected cat plasma was determined. In addition, IgG and IgA levels against the peptides were examined using plasma obtained during a time course following viral inoculation.

## 2. Results

### 2.1. Mapping of the B-Cell Linear Epitopes in FNoV VP1

To identify the B-cell linear epitopes in FNoV VP1, plasma samples from GVI FNoV M49-1-infected cats were screened by peptide-based ELISA (Figure 1). As shown in Appendix A, the amino acid sequence homology between GIV FNoV and GVI FNoV VP1 P domains was low (less than 45%). Therefore, plasma from GIV FNoV M81-infected cats was used to confirm the genogroup-specific reactivity of each peptide. The plasma samples of the GIV FNoV M49-1-infected cats were more reactive to three of the peptides (P-9, P-10, and P-11) than to the other peptides. Of note, the reactivity of GVI FNoV plasma samples to P-09 and P-10 was significantly higher (*p* < 0.01) than other peptides (Figure 1A). The plasma samples of GIV FNoV M81-infected cats and Specific pathogen-free (SPF) cats were not reactive to all peptides (Figure 1B,C). These results suggest that P-09, P-10, and P-11 contain B-cell linear epitopes.

### 2.2. Kinetics of IgG Response to Peptides during Course of Infection

To explore the kinetics of the IgG response to the peptides during the course of infection, P-09, P-10, and P-11 were evaluated by ELISA using plasma samples obtained from 5 GVI FNoV M49-1-infected cats at multiple time points (Figure 2A). An increase was observed in the IgG response to P-09, P-10, and P-11 over the time course. Antibodies to P-09 and P-10 were detected at 7 days post-infection (DPI), whereas antibodies to P-11 were not detected until 14 DPI. The IgG response to P-10 at 21 and 28 DPI was significantly higher than at 0 and 7 DPI. For P-09, a similar pattern was observed at 28 DPI.

### 2.3. Location of B-Cell Linear Epitope in FNoV VP1

The highest plasma IgG level for P-10 strongly supports that the B-cell linear epitope is within this peptide. Therefore, we mapped the site of the B-cell linear epitope in VP1 of GVI FNoV M49-1 (Figure 2B). The amino acid sequence encoded by P-10 was present in the loop structure of the P2 subdomain of VP1.

### 2.4. Viral Load, Clinical Symptoms, and Peptide-Specific Antibody Levels in Cats Infected with FNoV

We investigated whether the epitopes including P-10 are targets for a protective response to FNoV infection. We measured anti-peptide IgG (plasma P-10 IgG) levels in plasma samples, symptom scores, and viral load in stool samples from three FNoV-infected cats over a time course (Figure 3). Viral RNA was very high at 3–5 and 18–21 DPI, and this pattern of change in viral load was similar among all cats. Symptom scores were higher at 8–9 DPI. Plasma P-10 IgG levels were elevated with increasing viral load at 3–5 DPI; a decrease in symptom scoring was observed with increasing P-10 IgG levels. However, viral load increased again at 18–21 DPI even with high plasma P-10 IgG levels, suggesting that plasma P-10 IgG was not associated with FNoV proliferation in the late phase of infection. Based on this, we speculate that plasma P-10 IgA, but not IgG, decreases the viral load in the late stages of infection. Plasma P-10 IgA levels continued to increase from 7 to 28 DPI. In particular, two cats with relatively high plasma P-10 IgA levels at 28 DPI had low viral load levels in stool samples.

## 3. Discussion

In humans, it is not possible to conduct infection experiments using human volunteers with no history of HNoV infection for the purpose of elucidating the pathogenesis of the disease. Therefore, the study of human viral infections requires the use of animal models. Unfortunately, there is no suitable animal model that accurately reflects HNoV infection. Therefore, surrogate animal models are used to analyze HNoV infection in vivo. MNoV infection in mice is often used as a surrogate animal model for HNoV infection [25,26]. However, the disadvantages of this animal model include: (i) no clinical symptoms (no gastroenteritis observed) and (ii) viral shedding titers are lower than viral titers in the inoculum. Studies of HNoV infection using gnotobiotic animals have also been conducted [27,28]. These animals have immature immune systems and gastrointestinal tracts. Therefore, it is debatable whether gnotobiotic animals infected with HNoV reflect the actual pathology in humans. On the other hand, our FNoV model overcomes such disadvantages [29]. At present, the FNoV model is one of the best surrogate animal models for the study of HNoV infection.

There have been no previous reports on the B-cell epitopes in FNoV. Based on the results of this study that P-10 showed the strong antibody response, it was suggested that the B cell linear epitope (epitope A1) is located in aa 308–327 (location of P-10) on the P2 subdomain of VP1 of GVI FNoV. Structural analysis of GVI FNoV VP1 showed that the peptide containing epitope A1 formed a prominent loop region. Shanker et al. [20] reported that human IgA monoclonal antibody (IgA 5I2) with binding inhibitory activity against GI HNoV and HBGA recognize three loop structures (Loop T, Loop Q, and Loop U) exposed on the surface of the P2 subdomain of VP1.We speculate that the Loop containing P-10 is homologous to these Loop T. In addition, epitope D, one of the blockade epitopes of GII HNoV, is equivalent to a part of Loop T. Given this information, it is suggested that steric hindrance or allosteric HBGA blockade epitope of FNoV is likely to be present in P-10.

We investigated whether IgG and IgA against P-10 affect viral shedding and clinical symptoms. All three SPF cats experimentally infected with GVI FNoV showed an inverse correlation between plasma P-10 IgG and viral load in the early stages of infection. A similar relationship was found between plasma P-10 IgG and symptom score. However, a second viral shedding was observed in the late stage of infection even though plasma P-10 IgG levels were high. This indicates that plasma P-10 IgG was not associated with clearance against GVI FNoV infection in the late post-infection period. On the other hand, plasma P-10 IgA may be related to the inhibition of viral shedding. Future studies are needed to analyze the continuous relationship between anti-NoV IgG/IgA levels and viral load. Moreover, given the information in the studies of HNoV, there may be some antibodies that inhibit the shedding of the FNoV gene into the feces, in addition to antibodies targeting P-10. In the future, more detailed B-cell epitope analysis should be performed by producing monoclonal antibodies derived from VP1 of FNoV.

Few animal studies have examined the long-term and persistent shedding of viral genes in feces after norovirus infection; Thackray et al. [30]. reported the detection of viral genes in feces of MNoV-infected mice on day 35 after infection. However, viral genes in feces from day 7 to day 35 post-virus infection have not been investigated continuously. We collected fecal samples from FNoV-infected cats daily and continued to examine the number of viral genes in these samples, and confirmed that no viral genes were detected 7–10 days after viral infection. However, three weeks after viral infection, viral shedding was detected again. The reason for this is unknown; the MNoV CR6 strain is thought to persistently infect immune cells in lymphoid tissues of the gastrointestinal tract [30]. That is, if the immune cells are persistently infected by the virus, the virus can escape the immune effects of the host. The same phenomenon may have occurred in cats infected with FNoV.

Our study has several limitations. First, the FNoV model is a surrogate animal model, so the results of this study cannot be applied directly to the study of HNoVs. This is also true for other animal NoVs. Second, it is not clear whether antibodies against P-10 actually neutralize GVI FNoV. Third, unlike other laboratory animals, the number of cats available for experiments is low. In order to accelerate vaccine development for HNoV infection, a useful small animal model of the disease that accurately reflects the human condition is necessary.

There are few epidemiological studies of FNoV in veterinary medicine, despite FNoV being a potential pathogen of gastroenteritis in cats. Many veterinarians are unaware that FNoV is a potential pathogen of feline gastroenteritis. Considering that canine NoV was detected in 7.8% (21/268) of dogs with diarrhea symptoms [7], it is possible that the epidemiological situation in cats is similar to that in dogs.

## 4. Materials and Methods

### 4.1. Peptides

Twenty-four overlapping 20-mer peptides (offset every ten amino acids) spanning the P domains of GVI FNoV VP1 (equivalent to the GVI FNoV M49-1 strain; GenBank accession No. LC011950) were synthesized by Sigma-Aldrich Japan (Sigma-Aldrich Japan, Tokyo, Japan). The orientation and amino acid sequences of the peptides are shown in Figure 4. All peptides were dissolved in 50:50 acetonitrile:water at 1 mg/mL. Working aliquots were stored at −80 °C until use.

### 4.2. Plasma Samples for Screening of Peptides

Seven convalescent plasma samples from cats infected with GVI FNoV and 10 plasma samples from specific pathogen-free (SPF) cats were screened using peptide-based ELISA. Convalescent plasma samples were collected from cats experimentally infected with GVI FNoV M49-1 (n = 5) or GIV FNoV M81 (n = 2), as previously described [24,29,31].

### 4.3. Peptide-Based ELISA

Peptide-based ELISA was performed as previously described [32]. Briefly, Immulon 2HB (Thermo Fisher Scientific, Waltham, MA, USA) ELISA plates were coated with each peptide (add 100 µL of 1.0 µg/µL peptide solution to each well) overnight at 4 °C. After blocking with blocking buffer, 100 µL of plasma sample diluents (1:100) were applied onto the plates and incubated for 2 h at 37 °C. The bound antibodies were detected by HRP-conjugated goat anti-cat IgG or HRP-conjugated goat anti-cat IgA, followed by signal detection with *o*-phenylenediamine substrate solution. The reaction was stopped with stop solution and absorbance was measured by an ELISA reader.

### 4.4. Homology Modelling of FNoV VP1

Homology model building of VP1 of GVI FNoV M49-1 was performed according to the method of Qiao et al. [33]. Briefly, the crystal structure of VP1 of GII.2 HNoV (PDB accession code: 6OUC) was used as a template for the SWISS-MODEL homology-modelling server (https://swissmodel.expasy.org/, accessed on 13 May 2022).

### 4.5. Ethics of Animal Experiments

All applicable national and institutional guidelines for the care and use of animals were followed. The animal experimentation protocol was approved by the President of Kitasato University through the judgment of the Institutional Animal Care and Use Committee of Kitasato University (approval no. 18-104). SPF cats were maintained in a temperature-controlled isolation facility. Sample sizes were determined based on our experience with FNoV infection models, and the minimum number of cats was used.

### 4.6. Animal Infection

The experimental infection of GVI FNoV M49-1 in SPF cats was conducted using modified methods previously described [23]. Briefly, three one to three year old SPF cats were inoculated orally with 1 mL of GVI FNoV M49-1-gene-positive stool suspension (20%, *w*/*v*). Animals were monitored daily for body temperature and clinical signs. Stool samples were collected daily from each cat. Peripheral blood was collected every 7 days, and plasma was isolated. Symptom scoring of gastroenteritis in animals was performed following the method previously described [22]. In brief, the properties of the feces were scored by the three investigators as follows with reference to Liu et al. [34]. 0: normal feces; 1: mixed feces containing both solid and paste feces; 2: paste feces; 3: semi-liquid or liquid feces. Three points were added if the cat vomited. Total RNA was isolated from stool samples using a High Pure Viral RNA Isolation Kit (Roche Diagnostics, Indianapolis, IN, USA) following the manufacturer’s instructions. FNoV RNA level (viral load) in stool samples was quantified by quantitative reverse transcription PCR (qRT-PCR), which was performed according to our previous report [23].

### 4.7. Statistical Analyse

One-way ANOVA followed by Tukey-Kramer multiple comparison tests were performed for statistical analysis using JMP 16.2 (SAS, Cary, NC, USA).

## 5. Conclusions

Our results suggest that there is a linear epitope in the P2 subdomain of GVI FNoV VP1 that is recognized by antibodies involved in the inhibition of viral shedding and gastroenteritis-related symptoms. Induction of the antibody against loops of VP1 is common after FNoV infection, as demonstrated by HNoV infection. Antibodies against these loops inhibits binding of HBGAs and correlates with protection from HNoV infection. This study would be strengthened by a comparative study of the structural epitopes of HNoV with those of FNoV.

## Figures and Tables

**Figure 1 pathogens-11-00731-f001:**
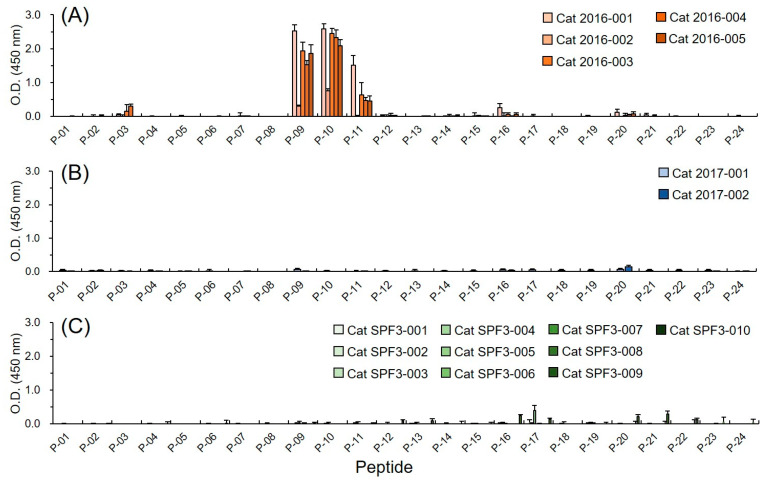
Reactivity of the peptides with plasma samples from FNoV-infected cats in indirect ELISA. Plasma (1:100) from cats infected with GVI FNoV M49-1 strain ((**A**) n = 5), GIV FNoV M81 strain ((**B**) n = 2), and specific pathogen-free (SPF) cats ((**C**) n = 10) were used to detect peptides including the B-cell linear epitope of the P domain of VP1. FNoV-infected plasma was collected 60 days after viral infection. HRP-labeled anti-cat IgG antibodies (1:1000) were used to detect antibodies bound to the peptide. The data represent the mean of three replicates.

**Figure 2 pathogens-11-00731-f002:**
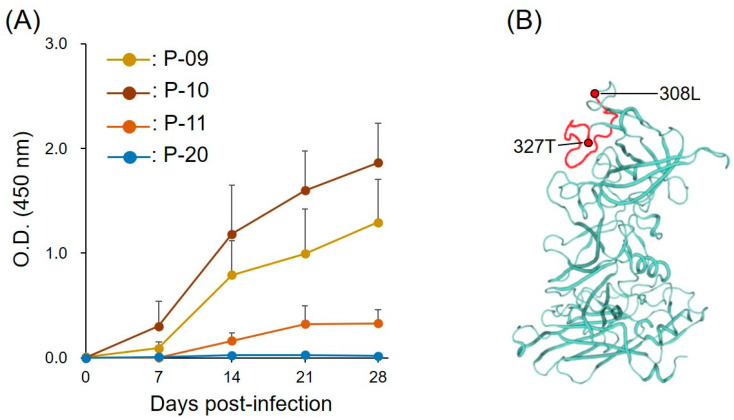
Recognition of peptides by plasma of FNoV-infected cats. (**A**) Time course of anti-peptide antibodies in plasma of FNoV-infected cats. Peptide-specific antibodies were detected in plasma once every 7 days from 0 DPI (the day of viral inoculation) to 28 DPI. The plasma samples used in the experiments were obtained from our previous studies [22]. Each mark indicates the mean value of the ELISA O.D. and the error bar indicates its standard error (mean value ± S.D.). (**B**) Homology modeling of GVI FNoV VP1 monomer. The position of P-10 within the P2 subdomain is colored red.

**Figure 3 pathogens-11-00731-f003:**
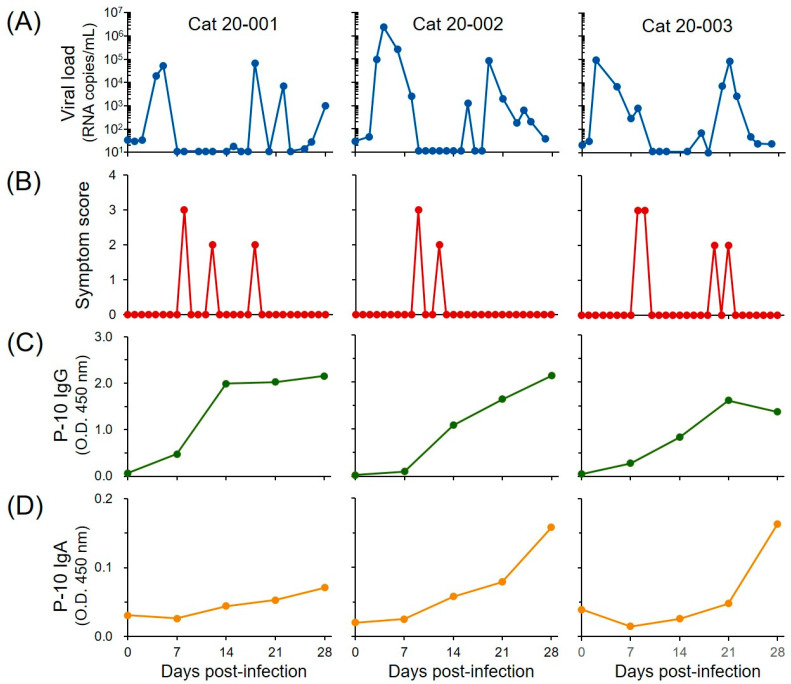
Viral shedding, clinical symptom score, and P-10 antibody response during the course of FNoV infection. (**A**) Viral RNA loads in the stool. (**B**) Clinical symptom scores. (**C**) Anti-P-10 IgG levels. (**D**) Anti-P-10 IgA levels.

**Figure 4 pathogens-11-00731-f004:**
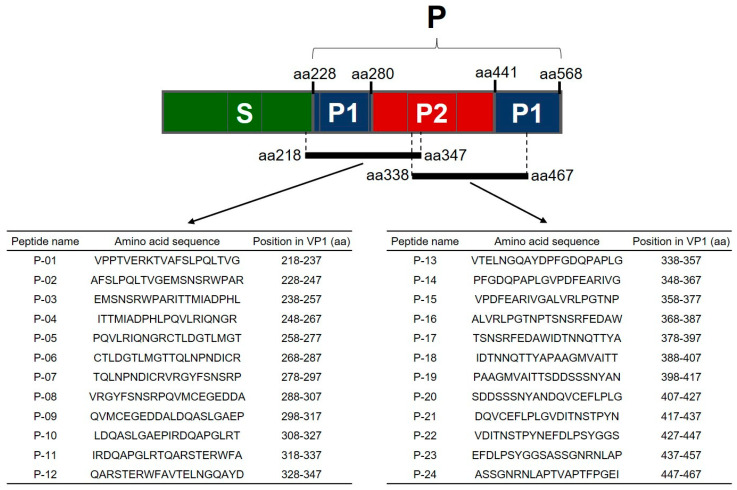
Schematic drawing of VP1 and peptide sequences derived from the P region of VP1.

## Data Availability

The data presented in this study are openly available in FigShare at https://doi.org/10.6084/m9.figshare.19904476.v1 (accessed on 13 May 2022) reference number 309341.

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
