# Peer review of "Immunodominant B-Cell Linear Epitope on the VP1 P Domain of a Feline Norovirus Cat Model"

_pathogens, 2022, doi:10.3390/pathogens11070731_

Round 1

Reviewer 1 Report

The study is fine but the manuscript has some issues needing to change before going on with the publication.

In this study Takano and colleagues used synthetic peptides, corresponding to the VP1 P domain of FnoV, to determine the immunoglobulin response against these peptides in the plasma of experimentally infected cats.

Although the experiments were correctly performed, some results are partial and many interpretations are flawed and biased.

Besides the manuscript was poorly written and badly formatted.

Minor concerns:

Lines 65-68: Not sure what you want to say in these sentences. Please rephrase it in order to deliver clearly your message

Lines 80-82: What kind of results are you trying to show in this table. You must describe these results besides the interpretation of the statistical analysis that was used.

Line 91 (Table 1): This table is useless. What is the meaning of these letters in the table?

Lines 92-99: Figure 1 needs a detailed description which is lacking in the text.

In addition, all elements in this figure must be described in the legend. For example, colored lines represent different peptides ...The 3D model is based on a previous structure of what viral strain?

Line 117: symptom scoring. Please give details on how symptoms are scaled

Line 125: This is figure 3, not figure 4

Line 143: Make clear you detected response against P-10, namely epitope linear A1

Lines 152-153: It is reasonable that immunoglobulins affect viral replication and mRNA levels but the data presented in figure 2 is not enough to conclude that IgA/IgG targets the P-10 (VP1) and has some effect to reduce viral loads.

Line 160: correct the typo (d in cats …)

Line 227: Conclusion must be placed after the discussion section.

Author Response

Response to the reviewer’s comments:

Reviewer #1:

The study is fine but the manuscript has some issues needing to change before going on with the publication. In this study Takano and colleagues used synthetic peptides, corresponding to the VP1 P domain of FnoV, to determine the immunoglobulin response against these peptides in the plasma of experimentally infected cats. Although the experiments were correctly performed, some results are partial and many interpretations are flawed and biased. Besides the manuscript was poorly written and badly formatted.

Response:

We sincerely thank the reviewer for taking time to consider our manuscript. 

Minor concerns:

Lines 65-68: Not sure what you want to say in these sentences. Please rephrase it in order to deliver clearly your message.

Response:

To reflect the reviewer’s comments, we modified the descriptions as;

“We previously reported that symptom scores and virus shedding in feces were inversely correlated with anti-recombinant partial VP1 (aa228-aa419) IgG levels in FNoV-infected cats [22]. Based on this fact, it is speculated that the immune response induced by the B-cell epitope in VP1 inhibits viral shedding and reduces symptoms of FNoV infection.” in Introduction (page 2, lines 70-73).

Lines 80-82: What kind of results are you trying to show in this table. You must describe these results besides the interpretation of the statistical analysis that was used.

Line 91 (Table 1): This table is useless. What is the meaning of these letters in the table?

Response:

Based on the opinions of all reviewers, Table.1 was removed from our manuscript. To reflect the reviewer’s comments, we modified the descriptions as;

“Of note, the reactivity of GVI FNoV plasma samples to P-09 and P-10 was significantly higher (p < 0.01) than other peptides (Figure. 1A).” in Results (page 2, lines 88-90).

Lines 92-99: Figure 1 needs a detailed description which is lacking in the text.

Response:

To reflect the reviewer’s comments, we modified the descriptions as;

“FNoV-infected plasma was collected 60 days after viral infection. HRP-labeled anti-cat IgG antibodies (1:1,000) were used to detect antibodies bound to the peptide.” in Figure 1 legend (page 3, lines 96-97).

In addition, all elements in this figure must be described in the legend. For example, colored lines represent different peptides ...The 3D model is based on a previous structure of what viral strain?

Response (Answer assuming Figure 2):

Thank you for pointing this out. We have added captions for each line in Figure 2A. The 3D model shown in Figure 2B is a homology modeling, which is stated in Figure 2 legend as;

“Homology modeling of GVI FNoV VP1 monomer.” (page 3, line 111).

Line 117: symptom scoring. Please give details on how symptoms are scaled

Response:

To reflect the reviewer’s comments, we modified the descriptions as;

“Symptom scoring of gastroenteritis in animals was performed following the method previously described [22]. In brief, the properties of the feces were scored by the three investigators as follows with reference to Liu et al. 0: normal feces; 1: mixed feces containing both solid and paste feces; 2: paste feces; 3: semi-liquid or liquid feces. Three points were added if the cat vomited.” in Materials and Methods (page 7, lines 240-244).

Line 125: This is figure 3, not figure 4

Response:

Thank you for pointing out our error. We have corrected it from “Figure 4” to “Figure 3” (page 4, line 121 and line 132).

Line 143: Make clear you detected response against P-10, namely epitope linear A1

Response:

To reflect the reviewer’s comments, we modified the descriptions as;

“Based on the results of this study that P-10 showed the strong antibody response, it was suggested that the B cell linear epitope (epitope A1) is located in aa 308-327 (location of P-10) on the P2 subdomain of VP1 of GVI FNoV.” in Discussion (page 5, lines 150-153).

We have corrected it from “epitope A” to “P-10” (page 5, line 162).

Lines 152-153: It is reasonable that immunoglobulins affect viral replication and mRNA levels but the data presented in figure 2 is not enough to conclude that IgA/IgG targets the P-10 (VP1) and has some effect to reduce viral loads.

Response:

Thank you again for the comments. To reflect the reviewer’s comments, we modified the descriptions as;

“Moreover, given the information in the studies of HNoV, there may be some antibodies that inhibit the shedding of the FNoV into the feces, in addition to antibodies targeting P-10. In the future, more detailed B-cell epitope analysis should be performed by producing monoclonal antibodies derived from VP1 of FNoV.” in Discussion (page 5, lines 171-174).

Line 160: correct the typo (d in cats …)

Response:

We have corrected the part you pointed out.

Reviewer 2 Report

In this manuscript Takano et al. define a B cell linear epitope in a feline calicivirus strain using plasma from infected cats.  They further investigate the correlation between plasma IgG and IgA titer and virus titer, symptoms, and clearance. 

The manuscript is well structured and interesting, but the data are not strong enough to support the some of their conclusions and there are numerous errors in data presentation.  In particular, the conclusion that plasma IgA may be correlated with viral clearance (not protection as stated in the text) is not supported, as little to no IgA to P-10 was detected in the plasma of three infected cats.  Since IgG to P-10 also did not correlate with virus clearance, the mechanism of clearance remains unknown.

In my opinion, the manuscript would be much stronger if the authors would focus on the FNoV they studied, which was generally well done (see exceptions below) and then discussed the potential commonalities between FNoV and human NoV that could be capitalized upon as an animal model that is more relevant than MNV infection in mice. 

Specific comments:

1.       The abstract needs to be modified in response the comments below.

2.       The manuscript is about feline NoV, yet FNoV is not mentioned in the introduction until the final paragraph.  Please include information about FNoV in the intro.

a.       Is the feline NoV genome structured like human NoV or MNV?

b.       Any information on disease symptoms, frequency, immunity to FNoV?

3.       Line 58 is misleading.  Many studies on mAb to human NoV have been published and epitopes mapped, including linear epitopes. 

4.       Line 69, do you mean HBGA-binding blocking antibody epitopes?  If so, how does that relate to the experiments you did as no Ab function beyond binding was measured?

5.       Line 82, do you mean GIV FNoV infected cat plasma were unreactive to all peptides (none reacted), as opposed to “not reactive to all peptides” (but reactive to some)?  Figure 1 supports GIV plasma was unreactive to all peptides.

6.       Line 83, define SPF.

7.       Table 1, is not necessary.  Stats on responses to peptides of interest between FNoV infected cats and SPF cats can be added in the text or Figure 4.

8.       Table 1, please define the letter codes used in the table.

9.       Figure 1A, the colored lines are not defined.  Which peptide is which line color?

10.   Line 103, add citation.

11.   Figure 1B, this is not a schematic arrangement.  It is a homology model.  Please correct the figure legend to match the correct text in the methods section.

12.   Can you compare the location of the P10 epitope to a human NoV epitope?  If so, is this epitope an HBGA binding blockade epitope?  This is would be a good discussion point.

13.   Line 121, the IgA OD is very low across all three cats.  How was the assay limit of detection determined?  Are OD <0.2 accurately quantifiable in your assay?

14.   Line 122, please explain how IgA levels at <0.2OD are considered “high plasma P-10 IgA”. 

15.   Figure 4A warrants further discussion, as all three cats appear to clear virus to levels below the LOD and then become positive again.  Please discuss if these findings are typical for FNoV infection and anything known about immunity during the 2 phases of virus shedding.  Please include in the discussion possible explanations for these observations.

16.   Figure 4, more information is needed on symptom scoring.

17.   There isn’t an obvious pattern between any combination of symptoms/titer/antibody.  Please discuss the observations and put them into context for the reader.

18.   Figure 4, were any uninfected cats monitored similarly to the cats in Figure 4?  It would help the reader interpret the responses reported in the infected cats, especially given the lack of consistency and patterns in the data sets.

19.   Linen 129, there have been multiple controlled human infection studies conducted in humans with human norovirus.  This sentence incorrect.

20.   Line 144, “epitope A1” nomenclature is only used in this paragraph and the following paragraph.  P-10 is used elsewhere before and after these two paragraphs.

21.    Line 148, for human norovirus, antibodies that block HBGA binding do not necessarily bind to the HBGA binding site.  Many block HBGA binding via steric hindrance and allosteric effects on particle conformation.  Please revise.

22.   Line 157, I believe you mean virus clearance not virus protection, since the animals have already been infected for more than a week.

23.   Figure 4, adding AA numbers to the linear schematic would help the reader localize the peptides in the domains.

24.   Line 192, please provide information on the IgA EIA.  Anti-IgG-HRP is listed as the only secondary antibody. 

25.   Line 195, “1ug/ul/well”, please clarify antigen coating.

26.   Line 216, what clinical symptoms?

27.   Line 232, no data in this manuscript confirms epitopes within loops of the P2 domain induce protective immunity against NoV infection.  Studies to measure protective immunity were not done.  Please revise this sentence.  The study did confirm Ab to loops are common post infection with FNoV as demonstrated with Human NoV infection.  Human Ab to these loops block binding of HBGAs and correlate with protection from infection (please cite appropriate papers).  The study would be strengthened with a comparison of structural epitopes in human NoV and how they compare to the FNoV epitope.

Author Response

Response to the reviewer’s comments:

Reviewer #2:

In this manuscript Takano et al. define a B cell linear epitope in a feline calicivirus strain using plasma from infected cats.  They further investigate the correlation between plasma IgG and IgA titer and virus titer, symptoms, and clearance. 

The manuscript is well structured and interesting, but the data are not strong enough to support the some of their conclusions and there are numerous errors in data presentation. In particular, the conclusion that plasma IgA may be correlated with viral clearance (not protection as stated in the text) is not supported, as little to no IgA to P-10 was detected in the plasma of three infected cats. Since IgG to P-10 also did not correlate with virus clearance, the mechanism of clearance remains unknown.

In my opinion, the manuscript would be much stronger if the authors would focus on the FNoV they studied, which was generally well done (see exceptions below) and then discussed the potential commonalities between FNoV and human NoV that could be capitalized upon as an animal model that is more relevant than MNV infection in mice.

Response:

Thank you for carefully reviewing our manuscript and providing thoughtful comments. Given the study of noroviruses, the kinetics of immunity, particularly in protection against infection and disease onset, may be of great value in gaining insight into the pathophysiology of norovirus infections. The present study provides new immunological information on feline norovirus infection as a potential surrogate animal model for human norovirus infection, highlighting the presence of a B-cell epitope. In addition, follow-up analysis using three cats provided basic evidence for our in vitro findings. As the reviewer points out, it is too early to conclude that P-10-induced IgA is important for protective immunity, but we hope that the reviewer will understand the scientific contribution of our study to the control of norovirus infection in humans and companion animals.

The abstract needs to be modified in response the comments below.

Response:

To reflect the reviewer’s comments, we modified the descriptions as;

“We compared antibody levels to peptides containing the B-cell linear epitope (P-10) in three FNoV-infected cats with time-course changes in viral load and symptom scoring. After FNoV infection, viral shedding and clinical symptoms were shown to improve by elevated levels of antibodies against P-10 in the plasma.” in Abstract (page 1, lines 14-17).

The manuscript is about feline NoV, yet FNoV is not mentioned in the introduction until the final paragraph. a. Please include information about FNoV in the intro Is the feline NoV genome structured like human NoV or MNV? b. Any information on disease symptoms, frequency, immunity to FNoV?

Response:

To reflect the reviewer’s comments, we modified the descriptions as;

“Genetically, GIV and GVI FNoVs are located more closely to human noroviruses than mouse noroviruses (MNoV: GV) and bovine noroviruses (GIII).” in Introduction (page 1, lines 42-44).

“We previously showed gastrointestinal symptoms in SPF cats orally challenged with FNoV gene-positive feces. This fact strongly suggests that FNoV is a pathogen that causes gastroenteritis in cats. At present, FNoV infection has not attracted much clinical attention because of its low symptoms. However, NoV has a higher rate of genetic mutation than other viruses. Therefore, it is possible that FNoV may mutate in the future to become a highly virulent strain.” in Introduction (page 2, lines 46-51).

Line 58 is misleading. Many studies on mAb to human NoV have been published and epitopes mapped, including linear epitopes.

Response:

To reflect the reviewer’s comments, we have removed the following text from the manuscript

“In the study of NoVs (except MNoV), the search for B-cell linear epitopes is difficult due to technical limitations of immunological analysis methods.”

Line 69, do you mean HBGA-binding blocking antibody epitopes?  If so, how does that relate to the experiments you did as no Ab function beyond binding was measured?

Response:

Thank you for pointing out. We have corrected from “HBGA-binding epitopes” to “B-cell linear epitopes” (page 2 line 75).

Line 82, do you mean GIV FNoV infected cat plasma were unreactive to all peptides (none reacted), as opposed to “not reactive to all peptides” (but reactive to some)?  Figure 1 supports GIV plasma was unreactive to all peptides.

Response:

As described by the reviewer, there were a few peptides that reacted slightly, but the majority did not react nearly as much to the peptide.

Line 83, define SPF.

Response:

Thank you for pointing out our error. We have corrected it from “SPF” to “Specific pathogen-free (SPF)” (page 2, lines 91-92).

Table 1, is not necessary. Stats on responses to peptides of interest between FNoV infected cats and SPF cats can be added in the text or Figure 4.

Table 1, please define the letter codes used in the table.

Response:

Based on the opinions of all reviewers, Table.1 was removed from our manuscript. To reflect the reviewer’s comments, we modified the descriptions as;

“Of note, the reactivity of GVI FNoV plasma samples to P-09 and P-10 was significantly higher (p < 0.01) than other peptides (Figure. 1A).” in Results (page 2, lines 88-90).

Figure 1A, the colored lines are not defined. Which peptide is which line color?

Response (Answer assuming Figure 2A):

Thank you for pointing this out. We have added captions for each line in Figure 2A.

Line 103, add citation

Response:

Thank you for pointing this out. I have cited as you indicated.

Figure 1B, this is not a schematic arrangement. It is a homology model. Please correct the figure legend to match the correct text in the methods section.

Response:

The 3D model shown in Figure 2B is a homology modeling, which is stated in Figure 2 legend as;

“Homology modeling of GVI FNoV VP1 monomer.” (page 3, line 111).

Can you compare the location of the P10 epitope to a human NoV epitope?  If so, is this epitope an HBGA binding blockade epitope?  This is would be a good discussion point

Response:

We agree with the reviewer. To reflect the reviewer’s comments, we added the descriptions as;

“Shanker et al. [20] reported that human IgA monoclonal antibody (IgA 5I2) with binding inhibitory activity against GI HNoV and HBGA recognize three loop structures (Loop T, Loop Q, and Loop U) exposed on the surface of the P2 subdomain of VP1.We speculate that the Loop containing P-10 is homologous to these Loop T. In addition, epitope D, one of the blockade epitopes of GII HNoV, is equivalent to a part of Loop T. Given these information, it is suggested that steric hindrance or allosteric HBGA blockade epitope of FNoV is likely to be present in P-10.” in Discussion (page 5, lines 154-161).

Line 121, the IgA OD is very low across all three cats.  How was the assay limit of detection determined?  Are OD <0.2 accurately quantifiable in your assay?

Line 122, please explain how IgA levels at <0.2OD are considered “high plasma P-10 IgA”.

Response:

In our preliminary experiments, the cutoff value of the ELISA for IgA detection was 0.071 (mean + 2 SD; set using 32 SPF cat plasma samples). Therefore, any OD value greater than 0.071 can be judged to be positive for anti P-10 IgA.

To reflect the reviewer’s comments, we modified the descriptions as;

“In particular, two cats with relatively high plasma P-10 IgA levels at 28 DPI had low viral load levels in stool samples.” in Results (page 4, lines 129-130).

Figure 4A warrants further discussion, as all three cats appear to clear virus to levels below the LOD and then become positive again.  Please discuss if these findings are typical for FNoV infection and anything known about immunity during the 2 phases of virus shedding.  Please include in the discussion possible explanations for these observations.

Response:

Thank you again for the comments. To reflect the reviewer’s comments, we added the descriptions as;

“Few animal studies have examined the long-term and persistent shedding of viral genes in feces after norovirus infection; Thackray et al. [32] reported the detection of viral genes in feces of MNoV-infected mice on day 35 after infection. However, viral genes in feces from day 7 to day 35 post-virus infection have not been investigated continuously. We collected fecal samples from FNoV-infected cats daily and continued to examine the amount of viral genes in these samples, and confirmed that no viral genes were detected 7-10 days after viral infection. However, 3 weeks after viral infection, viral shedding was detected again. The reason for this is unknown; MNoV CR6 strain is thought to persistently infect immune cells in lymphoid tissues of the gastrointestinal tract. That is, if the immune cells are persistently infected by the virus, the virus can escape the immune effects of the host. The same phenomenon may have occurred in cats infected with FNoV.” in Discussion (page 5, lines 175-185).

Figure 4, more information is needed on symptom scoring.

Response:

To reflect the reviewer’s comments, we modified the descriptions as;

“Symptom scoring of gastroenteritis in animals was performed following the method previously described [22]. In brief, the properties of the feces were scored by the three investigators as follows with reference to Liu et al. 0: normal feces; 1: mixed feces containing both solid and paste feces; 2: paste feces; 3: semi-liquid or liquid feces. Three points were added if the cat vomited.” in Materials and Methods (page 7, lines 240-244).

There isn’t an obvious pattern between any combination of symptoms/titer/antibody. Please discuss the observations and put them into context for the reader.

Response:

The style of the discussion was modified significantly in accordance with the suggestions of all reviewers.

Figure 4, were any uninfected cats monitored similarly to the cats in Figure 4?  It would help the reader interpret the responses reported in the infected cats, especially given the lack of consistency and patterns in the data sets.

Response:

I agree with the reviewer's point. However, the number of cats used is limited due to animal welfare, and we were unable to use cats under such conditions in this experiment. We hope you will understand this point.

Linen 129, there have been multiple controlled human infection studies conducted in humans with human norovirus. This sentence incorrect.

Response:

Thank you for pointing this out. To reflect the reviewer’s comments, we modified the descriptions as;

“In humans, it is not possible to conduct infection experiments using human volunteers with no history of HNoV infection for the purpose of elucidating the pathogenesis of the disease.” in Discussion (page 5, lines 136-138).

Line 144, “epitope A1” nomenclature is only used in this paragraph and the following paragraph. P-10 is used elsewhere before and after these two paragraphs.

Response:

To reflect the reviewer’s comments, we modified the descriptions as;

“Based on the results of this study that P-10 showed the strong antibody response, it was suggested that the B cell linear epitope (epitope A1) is located in aa 308-327 (location of P-10) on the P2 subdomain of VP1 of GVI FNoV.” in Discussion (page 5, lines 150-153).

We have corrected it from “epitope A” to “P-10” (page 5, line 162).

Line 148, for human norovirus, antibodies that block HBGA binding do not necessarily bind to the HBGA binding site. Many block HBGA binding via steric hindrance and allosteric effects on particle conformation. Please revise.

Response:

We agree with the reviewer. To reflect the reviewer’s comments, we added the descriptions as;

“Shanker et al. [20] reported that human IgA monoclonal antibody (IgA 5I2) with binding inhibitory activity against GI HNoV and HBGA recognize three loop structures (Loop T, Loop Q, and Loop U) exposed on the surface of the P2 subdomain of VP1.We speculate that the Loop containing P-10 is homologous to these Loop T. In addition, epitope D, one of the blockade epitopes of GII HNoV, is equivalent to a part of Loop T. Given these information, it is suggested that steric hindrance or allosteric HBGA blockade epitope of FNoV is likely to be present in P-10.” in Discussion (page 5, lines 154-161).

To reflect the reviewer’s comments, we have removed the following text from the manuscript;

“Antibodies that react with the loop region of the HNoV VP1 P2 subdomain prevent virion binding to HBGA [20]. Based on this, we speculated that epitope A1 has the HBGA binding site(s). In the future, VLPs of GVI FNoV should be generated to determine whether epitope A1 is important for HBGA binding.”

Line 157, I believe you mean virus clearance not virus protection, since the animals have already been infected for more than a week.

Response:

Thank you for pointing out our error. We have corrected it from “protection” to “clearance” (page 5, lines 167-168).

Figure 4, adding AA numbers to the linear schematic would help the reader localize the peptides in the domains.

Response:

We thank you for your constructive comments on our paper and have included the AA number in Figure 4.

Line 192, please provide information on the IgA EIA. Anti-IgG-HRP is listed as the only secondary antibody.

Response:

To reflect the reviewer’s comments, we modified the descriptions as;

“The bound antibodies were detected by HRP-conjugated goat anti-cat IgG or HRP-conjugated goat anti-cat IgA, followed by signal detection with o-phenylenediamine substrate solution.” in Materials and Methods (pages 6-7, lines 217-220).

Line 195, “1ug/ul/well”, please clarify antigen coating.

Response:

To reflect the reviewer’s comments, we modified the descriptions as;

“Briefly, Immulon 2HB (Thermo Fisher Scientific, MA, USA) ELISA plates were coated with each peptide (1.0 µg/µL/well) overnight at 4°C.” in Materials and Methods (page 6, lines 214-216).

Line 216, what clinical symptoms?

Response:

To reflect the reviewer’s comments, we modified the descriptions as;

“Symptom scoring of gastroenteritis in animals was performed following the method previously described [22]. In brief, the properties of the feces were scored by the three investigators as follows with reference to Liu et al. 0: normal feces; 1: mixed feces containing both solid and paste feces; 2: paste feces; 3: semi-liquid or liquid feces. Three points were added if the cat vomited.” in Materials and Methods (page 7, lines 240-244).

Line 232, no data in this manuscript confirms epitopes within loops of the P2 domain induce protective immunity against NoV infection.  Studies to measure protective immunity were not done.  Please revise this sentence.  The study did confirm Ab to loops are common post infection with FNoV as demonstrated with Human NoV infection.  Human Ab to these loops block binding of HBGAs and correlate with protection from infection (please cite appropriate papers).  The study would be strengthened with a comparison of structural epitopes in human NoV and how they compare to the FNoV epitope.

Response:

Thank you for the kind advice. To reflect the reviewer’s comments, we modified the descriptions as;

“Induction of the antibody against loops of VP1 is common after FNoV infection, as demonstrated by HNoV infection. Antibodies against these loops inhibits binding of HBGAs and correlates with protection from HNoV infection. This study would be strengthened by a comparative study of the structural epitopes of HNoV with those of FNoV.” in Conclusion (page 7, lines 255-259).

Reviewer 3 Report

Takano et al. report an immunodominant linear epitope mapping on the P domain from a feline norovirus. Particularly, using the sera from animals infected with this feline norovirus they showed that the presence of antibodies targeting this immunodominant epitope correlated with lower virus shedding and clinical symptoms. This information is very relevant not only for feline norovirus research, but also to validate their animal model for human norovirus research. Notably, the epitope defined here (aa 308-327) maps on the same loop that epitope A from GII.4 noroviruses. Epitope A from GII.4 noroviruses has been recently shown to be immunodominant and involved in neutralization of human noroviruses (Tohma et al. Cell Reports 2022). Together, I believe this is important information for norovirus vaccine research. I commend this group for the development of this animal model and I encourage them to continue their work.

Specific comments:

1. Lines 36-37: VP1 is not the most important protein for norovirus infectivity. Infection is the result of the orchestrated action of multiple viral proteins. Please considering rephrasing. 

2. Lines 47-48: The group of Stephanie Karst recently developed a mice model that seems to present gastrointestinal symptoms. Please consider citing (Roth et al. Nature Communications 2020) or rephrasing the statement.

3. Lines 60-63: Please provide a more exhaustive literature reference for the report of B-cell linear epitopes described for human noroviruses. There are way over two manuscripts reporting linear epitopes mapping on VP1. Please omit reference #21. This is an in silico analysis without any experimental validation and therefore can lead to misinformation.

4. Lines 77-80: It was difficult to this reviewer to understand the origin of M81-infected cats and that this virus belongs to a different genogroup. This virus was not reported except in the recent genomic analyses done for GIV and GVI noroviruses (Ford-Siltz et al. Viruses 2019). Please provide additional background information about the diversity of feline noroviruses in the introduction section and consider presenting a table/figure that shows the genetic differences between these two viruses on the reported epitope (aa 308-327). The latter will provide information on the degree of mutations required to distinguish norovirus genotypes/genogroups.

5. I do not see the need of Table 1. All these data is clearly shown in Figure 1. Therefore, please consider omitting this table.

6. Please provide a legend for the colored-lines from Figure 2 Panel A.

7. It is not clear why the authors used the crystal structure of a human GII.2 (PDB: 6OUC) for modeling, instead of using the one available for GIV.2 feline norovirus (PDB= 4QUZ). Please use the latter structure and include the following reference: Singh et al. Virology 2015 474: 181-185.

8. Your speculation that plasma IgA directed to epitope aa 308-327, but not IgG, decreases the viral load is based on data from 1-2/3 animals. While you mentioned that further studies are needed to confirm this observation, I will suggest to tone down this claim until additional information is available. Why only 3/5 animals were included in the analyses from Figure 4?

9. In one of your limitations you claim "it is not clear whether antibodies against P-10 actually neutralize GVI" feline norovirus; however, you fail to comment that the homologous site (Epitope A) on the structure of GII.4 norovirus has been shown to be involved in neutralization of human noroviruses (Tohma et al. Cell Reports 2022). Please make a better argument on why your animal model and findings is very relevant for human norovirus studies. In light of these new data on neutralization from human noroviruses, the statements from lines 146-148 should be modified as they are outdated.

10. Please change "blood anti-norovirus IgA" by "systemic anti-norovirus IgA" or "anti-norovirus IgA circulating in blood"

Author Response

Response to the reviewer’s comments:

Reviewer #3

Takano et al. report an immunodominant linear epitope mapping on the P domain from a feline norovirus. Particularly, using the sera from animals infected with this feline norovirus they showed that the presence of antibodies targeting this immunodominant epitope correlated with lower virus shedding and clinical symptoms. This information is very relevant not only for feline norovirus research, but also to validate their animal model for human norovirus research. Notably, the epitope defined here (aa 308-327) maps on the same loop that epitope A from GII.4 noroviruses. Epitope A from GII.4 noroviruses has been recently shown to be immunodominant and involved in neutralization of human noroviruses (Tohma et al. Cell Reports 2022). Together, I believe this is important information for norovirus vaccine research. I commend this group for the development of this animal model and I encourage them to continue their work.

Response:

We truly thank the reviewer for careful review of our manuscript and kindly acknowledging the value of our study. We appreciate many thoughtful comments, which substantially contributed to further improvement of the manuscript.

Specific comments:

Lines 36-37: VP1 is not the most important protein for norovirus infectivity. Infection is the result of the orchestrated action of multiple viral proteins. Please considering rephrasing.

Response:

Thank you for the kind advice. To reflect the reviewer’s comments, we modified the descriptions as;

“VP1 can be divided into two major regions: the shell (S) domain and the protruding (P) domain [10].” in Introduction (page 1, lines 35-36).

Lines 47-48: The group of Stephanie Karst recently developed a mice model that seems to present gastrointestinal symptoms. Please consider citing (Roth et al. Nature Communications 2020) or rephrasing the statement.

Response:

To reflect the reviewer’s comments, we modified the descriptions as;

“There are no reports of non-human animals presenting gastroenteritis due to HNoV infection, except for wild-type neonatal mouse and gnotobiotic animals that have poorly developed immunity [13,33]. Adult mice and pigs do not develop gastroenteritis after inoculation with host-specific NoVs.” in Introduction (page 2, lines 53-56).

Lines 60-63: Please provide a more exhaustive literature reference for the report of B-cell linear epitopes described for human noroviruses. There are way over two manuscripts reporting linear epitopes mapping on VP1. Please omit reference #21. This is an in silico analysis without any experimental validation and therefore can lead to misinformation.

Response:

To reflect the reviewer’s comments, we have removed the following text from the manuscript

“Comprehensive in silico detection of the B-cell linear epitope of HNoV has also been reported [21].”

We cited several papers that studied the B-cell epitope of norovirus.

Lines 77-80: It was difficult to this reviewer to understand the origin of M81-infected cats and that this virus belongs to a different genogroup. This virus was not reported except in the recent genomic analyses done for GIV and GVI noroviruses (Ford-Siltz et al. Viruses 2019). Please provide additional background information about the diversity of feline noroviruses in the introduction section and consider presenting a table/figure that shows the genetic differences between these two viruses on the reported epitope (aa 308-327). The latter will provide information on the degree of mutations required to distinguish norovirus genotypes/genogroups.

Response:

Based on the opinions of all reviewers, Figure 1 was removed from our manuscript. To reflect the reviewer’s comments, we modified the descriptions as;

“Genetically, GIV and GVI FNoVs are located more closely to human noroviruses than mouse noroviruses (MNoV: GV) and bovine noroviruses (GIII). GIV FNoV has been identified in cats in the U.S. and Japan, and GVI FNoV in cats in Japan and Italy.” in Introduction (page 1, lines 42-45).

“As shown in Supplementary Figure S1, the amino acid sequence homology between GIV FNoV and GVI FNoV VP1 P domains was low (less than 45%). Therefore, plasma from GIV FNoV M81-infected cats was used to confirm the genogroup-specific reactivity of each peptide.” in Results (page 2, lines 83-86).

I do not see the need of Table 1. All these data is clearly shown in Figure 1. Therefore, please consider omitting this table.

Response:

Based on the opinions of all reviewers, Table.1 was removed from our manuscript. To reflect the reviewer’s comments, we modified the descriptions as;

“Of note, the reactivity of GVI FNoV plasma samples to P-09 and P-10 was significantly higher (p < 0.01) than other peptides (Figure. 1A).” in Results (page 2, lines 88-90).

Please provide a legend for the colored-lines from Figure 2 Panel A.

Response:

Thank you for pointing this out. We have added captions for each line in Figure 2A.

It is not clear why the authors used the crystal structure of a human GII.2 (PDB: 6OUC) for modeling, instead of using the one available for GIV.2 feline norovirus (PDB= 4QUZ). Please use the latter structure and include the following reference: Singh et al. Virology 2015 474: 181-185.

Response:

Thank you for your insightful suggestions and advice. We tried to perform homology modeling based on the crystal structure of the VP1 P domain of GIV.2 feline norovirus (4QUZ). Unfortunately, the PDB data of 4QUZ was missing the region containing P-10 (See figure below). Therefore, no changes were made to the homology modeling based on 6OUC.

Figure. GIV.2 Crystal structure of GIV.2 FNoV (4QUZ).

  • Please see attached word file.

Your speculation that plasma IgA directed to epitope aa 308-327, but not IgG, decreases the viral load is based on data from 1-2/3 animals. While you mentioned that further studies are needed to confirm this observation, I will suggest to tone down this claim until additional information is available. Why only 3/5 animals were included in the analyses from Figure 4?

Response:

To reflect the reviewer’s comments, we have removed the following text from the manuscript

“Interestingly, the second viral shedding was reduced in cats with increased plasma P-10 IgA levels at 28 DPI. It has been reported that systemic anti-norovirus IgA levels are important for protection against NoV infection [31]. However, to our knowledge, there are no reports comparing systemic anti-norovirus IgA levels and viral load continuously.”

To reflect the reviewer’s comments, we modified the descriptions as;

“Future studies are needed to analyze the continuous relationship between anti-NoV IgG/IgA levels and viral load.” in Discussion (page 5, lines 169-171).

In one of your limitations you claim "it is not clear whether antibodies against P-10 actually neutralize GVI" feline norovirus; however, you fail to comment that the homologous site (Epitope A) on the structure of GII.4 norovirus has been shown to be involved in neutralization of human noroviruses (Tohma et al. Cell Reports 2022). Please make a better argument on why your animal model and findings is very relevant for human norovirus studies. In light of these new data on neutralization from human noroviruses, the statements from lines 146-148 should be modified as they are outdated.

Response:

We sincerely thank the reviewers for their advice, and we have added a new claim to our discussion regarding the relationship between P-10 and previously known HNoV epitopes;

 “Shanker et al. [20] reported that human IgA monoclonal antibody (IgA 5I2) with binding inhibitory activity against GI HNoV and HBGA recognize three loop structures (Loop T, Loop Q, and Loop U) exposed on the surface of the P2 subdomain of VP1.We speculate that the Loop containing P-10 is homologous to these Loop T. In addition, epitope D, one of the blockade epitopes of GII HNoV, is equivalent to a part of Loop T. Given these information, it is suggested that steric hindrance or allosteric HBGA blockade epitope of FNoV is likely to be present in P-10.” in Discussion (page 5, lines 154-161).

Please change "blood anti-norovirus IgA" by "systemic anti-norovirus IgA" or "anti-norovirus IgA circulating in blood"

Response:

Thank you for pointing out. The points mentioned by the reviewers have been removed from the manuscript through revisions based on other comments.

Round 2

Reviewer 1 Report

In this new document, the authors provided an improved version of the manuscript. There are no additional issues with the current manuscript.

Author Response

In this new document, the authors provided an improved version of the manuscript. There are no additional issues with the current manuscript.

Response: The quality of our papers has been improved by the reviewers' sincere suggestions. We would like to express our sincere appreciation.

Reviewer 2 Report

Figure 2A: define marker and error bars.

Methods: (1.0 μ g/μ L/well) still does not define how much antigen is in the well unless you also provide the volume,unless you are saying 1ul went into the well.

Author Response

We sincerely thank you again for your review of our paper. We have included our response below.

Figure 2A: define marker and error bars.

Response: We thank the reviewer for your suggestions. The following text has been added to the Figure 2 legend; "Each mark indicates the mean value of the ELISA O.D. and the error bar indicates its standard error (mean value ±S.D.). "

Methods: (1.0 μ g/μ L/well) still does not define how much antigen is in the well unless you also provide the volume,unless you are saying 1ul went into the well.

Response: We apologize for pointing this out again. We have corrected the part you pointed out as follows; "(add 100µL of 1.0µg/µL peptide solution to each well)".